# Butyrate and the Intestinal Epithelium: Modulation of Proliferation and Inflammation in Homeostasis and Disease

**DOI:** 10.3390/cells10071775

**Published:** 2021-07-14

**Authors:** Pooja S. Salvi, Robert A. Cowles

**Affiliations:** Department of Surgery, Division of Pediatric Surgery, Yale School of Medicine, Yale University, New Haven, CT 065110, USA; pooja.a.shah@yale.edu

**Keywords:** microbiome, short chain fatty acid, butyrate, intestinal epithelium

## Abstract

The microbial metabolite butyrate serves as a link between the intestinal microbiome and epithelium. The monocarboxylate transporters MCT1 and SMCT1 are the predominant means of butyrate transport from the intestinal lumen to epithelial cytoplasm, where the molecule undergoes rapid β-oxidation to generate cellular fuel. However, not all epithelial cells metabolize butyrate equally. Undifferentiated colonocytes, including neoplastic cells and intestinal stem cells at the epithelial crypt base preferentially utilize glucose over butyrate for cellular fuel. This divergent metabolic conditioning is central to the phenomenon known as “butyrate paradox”, in which butyrate induces contradictory effects on epithelial proliferation in undifferentiated and differentiated colonocytes. There is evidence that accumulation of butyrate in epithelial cells results in histone modification and altered transcriptional activation that halts cell cycle progression. This manifests in the apparent protective effect of butyrate against colonic neoplasia. A corollary to this process is butyrate-induced inhibition of intestinal stem cells. Yet, emerging research has illustrated that the evolution of the crypt, along with butyrate-producing bacteria in the intestine, serve to protect crypt base stem cells from butyrate’s anti-proliferative effects. Butyrate also regulates epithelial inflammation and tolerance to antigens, through production of anti-inflammatory cytokines and induction of tolerogenic dendritic cells. The role of butyrate in the pathogenesis and treatment of intestinal neoplasia, inflammatory bowel disease and malabsorptive states is evolving, and holds promise for the potential translation of butyrate’s cellular function into clinical therapies.

## 1. Introduction

Biological understanding of the complex and dynamic homeostatic mechanisms of the intestinal mucosa has recently become even more intricate, as investigators begin to consider the role of the microbiome in health and disease. In this review, we focus on the microbial metabolite butyrate, which represents one of the most extensively researched molecular mediators in the host–microbiome relationship. Produced by colonic bacteria, butyrate modulates several epithelial processes throughout the gastrointestinal tract, including cell cycle progression, inflammation, and barrier integrity. Studies to date suggest that butyrate is a chameleon in its ability to exert distinct, but sometimes contradictory effects on epithelial cells under different conditions. Evolving knowledge of the intestinal crypt at a molecular level has now permitted a more detailed and discriminating appreciation for these incongruous experimental results. Understanding how butyrate interacts differently with differentiated enterocytes as opposed to undifferentiated cells in the epithelial crypt is of value for scientists and clinicians interested in intestinal neoplasms, inflammatory bowel disease, and malabsorptive states.

## 2. Production and Fate of Butyrate in the Intestinal Lumen

Several species of commensal Gram-positive bacteria in the colon possess the ability to synthesize butyrate, primarily from dietary starch and fiber, with the two most abundant groups appearing to be *Faecalibacterium prausnitzii* and *Roseburia* species, belonging to clostridial clusters IV and XIVa, respectively [1,2]. Bacterial synthesis of butyrate involves formation of butyryl coenzyme A from two molecules of acetyl coenzyme A, followed by conversion to butyrate by one of two known enzymatic pathways (reviewed by Louis et al. [2]).

Butyrate is structurally classified as a short-chain fatty acid (SCFA), along with the similar molecules, acetate and propionate. The luminal concentration of butyrate in humans and animals has been estimated at 10–20 mM, which is higher than other SCFA [3,4]. Ninety-five percent of butyrate in the colon is absorbed by colonocytes, for which it serves as a dominant energy source via β-oxidation and the tricarboxylic acid cycle [5]. While small intestinal enterocytes can also absorb butyrate, these cells primarily derive energy from glucose and glutamate [6].

At physiologic pH, butyrate exists in its ionized form, and thus requires carrier-mediated entry across the epithelial cell membrane. MCT1, a member of the proton-linked monocarboxylate transporter group is an apically oriented membrane-bound protein that is a major channel for intracellular SCFA transport [7]. Additional bicarbonate and sodium-linked transport proteins, including SMCT1, mediate passage of butyrate across both the apical and basolateral membranes, to a lesser extent (reviewed by Cook [8]). Systemic absorption of butyrate is low, as the majority is rapidly metabolized for fuel by epithelial cells. When absorbed, SCFAs travel via the portal circulation to the liver, where they are metabolized for energy or serve as substrate for longer-chain fatty acids [3].

## 3. Butyrate and Intestinal Epithelial Proliferation

### 3.1. The Butyrate Paradox

Multiple studies have demonstrated an association between a high-fiber diet and decreased risk of colon cancer [9,10,11]. These data naturally led into investigation of butyrate’s effects on colonocyte division, based on the link between dietary fiber and butyrate production [12]. Whitehead et al. were among the first to demonstrate that butyrate slowed proliferation and induced differentiation in a colon cancer cell line [13]. These findings were subsequently replicated in various tumor-derived cell lines [14,15,16]. However, animal models failed to show equivalent effects on the intestinal epithelium in vivo, and in fact showed enhanced epithelial proliferation in response to butyrate [17,18,19]. Furthermore, experiments with non-tumor cell lines showed that butyrate stimulated proliferation and inhibited differentiation in culture. This phenomenon of the molecule’s opposing effects on healthy versus cancerous colonocytes was dubbed the butyrate paradox [20,21].

An explanation for the butyrate paradox may be found in epigenetics. Butyrate has long been known to modify chromatin structure through histone modification [22]. Whitlock is credited with first linking the histone deacetylase (HDAC) inhibitor properties of butyrate with cell cycle inhibition [23]. In differentiated intestinal epithelial cells, butyrate is rapidly metabolized for fuel and thus does not have the opportunity to inhibit HDAC. In contrast, colon cancer cells preferentially use glucose, rather than SCFA, as fuel for cellular function—an adaptation that facilitates survival of neoplastic cells, referred to as the Warburg effect. The dominance of glycolytic metabolism over oxidative phosphorylation in cancerous colonocytes allows butyrate to accumulate and act as an HDAC inhibitor, thereby halting cell cycle progression through altered gene expression (see Figure 1) [24]. There is growing evidence that butyrate also directly binds and alters activity of metabolic enzymes in colon cancer cells, thus conferring a protective anti-neoplastic effect through reversal of the Warburg effect [25,26,27].

### 3.2. Transcriptional Activation of Cell Cycle Genes Offer Protection from Colon Cancer

Butyrate-induced cell cycle arrest occurs predominantly in G1 phase, with lower levels of inhibition also observed in G2 and M phases [28,29]. Chromatin analysis has revealed that butyrate results in Histone 3 hyperacetylation with subsequent enrichment of important cell cycle regulatory proteins including cyclin D1 and p21 [30,31]. Importantly, cell cycle inhibition is accompanied by an increase in cellular differentiation, which is key in counteracting uncontrolled proliferation by tumor cells [32,33].

Daly and colleagues demonstrated that MCT1-dependent transport of butyrate across the cell membrane is linked to the differential gene expression described above [34]. Others have similarly suggested that the SMCT1 transporter functions as a tumor-suppressor gene—in the colon and in other tissues—due to its importance in maintaining the intracellular butyrate concentration necessary for gene expression associated with cell cycle regulation [35,36]. As such, intracellular butyrate transport may be an important prognostic consideration in colon cancer.

Transcriptional activation of proteins involved in cellular apoptosis also contributes to the anti-neoplastic effects of butyrate in colon cancer. Butyrate-induced expression of numerous pro-apoptotic genes, such as *Bax* and *Bak*, and suppression of anti-apoptotic genes such as *Bcl-2*, lead to induction of the caspase cascade in tumor cells [37,38]. Notably, expression of apoptotic genes was not reduced in cells exhibiting downregulation or absence of MCT1 [34]. The low intracellular concentrations of butyrate in these cells suggests that regulation of colonocyte apoptosis occurs independently of histone modification. Rather, the proposed mechanism involves binding of butyrate to GPR109a, a G-protein coupled receptor (GPCR) present on the apical membrane of intestinal epithelial cells [39]. GPR109a binding exerts a tumor suppressive effect through reduced expression of *Bcl-2* and *Bcl-xL*, as well as through regulation of the Wnt/β-catenin signaling pathway that is fundamental to colorectal carcinogenesis [39,40].

Though considerable data supports the antineoplastic potential of butyrate, clinical application of what are largely cell culture-based experimental findings has yet to be established. Progress toward therapeutic translation is being made, however, with in vivo studies demonstrating the success of HDAC inhibitors in preventing and treating malignancies, including colorectal and prostate cancers [41,42]. Furthermore, microbiome characterization using 16S rRNA sequencing has emerged as a tool for identifying specific microbial signatures associated with colon cancer, which may aid in risk stratification [43,44].

### 3.3. Butyrate and the Crypt as a Lens into Microbiome–Host Coevolution

The intestinal crypt is the proliferative compartment of the epithelium, composed of intestinal stem cells and rapidly dividing transit amplifying cells [45]. As such, disruption of the cell cycle or cellular division in the crypt compartment has significant consequences for the entirety of the intestinal epithelium. There is evidence that cells at the crypt base rely predominantly on glycolysis over the citrate cycle and oxidative phosphorylation for cellular fuel—similar to neoplastic cells—and thus are susceptible to the downstream effects of intracellular butyrate accumulation [46,47].

Kaiko et al. recently presented evidence that the crypt morphology in humans and higher order mammals may have evolved as a means to protect intestinal stem cells and proliferating cells from the inhibitory effects of butyrate [46]. First, they illustrated that rapid uptake and oxidation of butyrate by surface-level colonocytes decreases butyrate levels at the crypt base and produces a butyrate gradient along the crypt axis in mouse colon. In contrast, the evolutionarily primitive zebrafish lack butyrate-producing bacterial organisms, as well as intestinal crypts. Second, the investigators demonstrated that exposure of intestinal stem cells to butyrate resulted in HDAC inhibition-mediated expression of Foxo3, a transcription factor that regulates cell cycle arrest and apoptosis through disruption of cellular metabolism [48]. When butyrate comes in contact with intestinal stem cells, such as under conditions of mucosal injury in crypt-containing mice, or from luminal infusion in crypt-less zebrafish, there is resulting inhibition of stem cell proliferation (see Figure 2).

Notably, Foxo3 upregulation has not been specifically observed in colon cancer cells treated with butyrate, although Foxo3 expression does modulate a number of pro-apoptotic genes associated with butyrate’s antineoplastic effects [48]. The importance of this transcription factor in intestinal stem cell cycle arrest, and the similarities in cellular metabolism between stem cells and neoplastic cells, call for further characterization of the antiproliferative pathways activated by butyrate in cancer cells.

Others have highlighted the nuanced interaction between butyrate at different levels of the crypt, as well. For example, butyrate induces expression of heat shock proteins (HSP) in intestinal epithelial cells, which interfere with the Wnt-signaling pathway necessary to maintain the undifferentiated state of intestinal stem cells [49]. Utilization of butyrate by cells at the top of the crypt results in insufficient levels for HSP induction at the base, and thus allows for unhindered Wnt activation and preservation of multipotent intestinal stem cells. These data offer insight into how the intestinal epithelium has evolved to tolerate—and benefit from—colonization by butyrate-producing microbial species.

### 3.4. Small Intestinal Enterocytes versus Colonocytes

The vast majority of data on butyrate’s regulation of intestinal epithelial proliferation come from experiments with colonocytes. Despite the obvious relation and general similarities between the small and large intestine, key differences in epithelial morphology and function make it difficult to draw unfettered parallels between the two [50]. Existing research into the proliferative effects of butyrate in the small intestine suggests that it has a role in epithelial homeostasis, but it is insufficient to judge whether this interaction is distinct from that in the colon. The butyrate receptors MCT1 and SMCT1 are present on small intestinal epithelial cells [51,52,53]. While these cells have the capacity to oxidize butyrate for cellular energy in times of starvation, glutamate and glucose are their preferred fuel sources [6].

In vitro, butyrate inhibits proliferation of immortalized small intestinal epithelial cells [54]. However, it has been suggested that the role of butyrate in cell culture can be influenced by the presence of alternate fuel sources—such as glucose—in cell media, and thus should be interpreted accordingly [55].

In animal studies, increasing luminal butyrate concentration has resulted in higher small intestinal crypt proliferation rates, whereas butyrate-deprivation has been associated with increased expression of pro-apoptotic proteins in small intestinal epithelial cells [56]. However, it is unclear whether these changes were observed in differentiated or undifferentiated epithelial cells. The degree of cellular differentiation is an important factor in predicting the downstream effects of butyrate on colonocytes, as described above, and may have bearing on its interaction with the small intestinal epithelium as well. Since regulation of intestinal proliferation is relevant to the management of malabsorptive and inflammatory intestinal diseases, which have been associated with microbial dysbiosis, further investigation of butyrate’s role in the small intestinal epithelium is warranted [57].

## 4. Butyrate as an Immunomodulator in the Intestine and Beyond

The delicate balance between tolerance of commensal gut bacteria and inflammation or injury in the setting of altered epithelial homeostasis is a topic of immense scientific interest, as it has the potential to elucidate the ways in which the environment, intestinal epithelium, and immune system converge to manifest disease. Butyrate and other SCFA have been identified at the center of this complex interaction. Specifically, butyrate-producing bacteria and intestinal butyrate concentration are associated with both local inflammation, as in the case of inflammatory bowel disease (IBD) or food allergy, and extraintestinal immunologic activity, including neuroinflammation.

### 4.1. Butyrate’s Role in Attenuating Intestinal Inflammation

There is strong evidence that the pathogenesis of intestinal inflammation, for example in inflammatory bowel disease (IBD) and food allergy, involve dysregulation of the epithelial immune response to the intestinal microbiome [58,59,60]. In particular, data showing depletion of butyrate-producing bacteria in IBD patients, and lower intestinal concentrations of butyrate in IBD patients compared to healthy counterparts drew attention to the potential immunomodulatory role of butyrate [61,62,63].

It was previously hypothesized that barriers to butyrate oxidation predisposed individuals to epithelial injury, which thus led to intestinal inflammation [64]. Based on this theory, butyrate supplementation was proposed as a treatment to attenuate IBD severity [65,66]. However, further research demonstrated that non-inflamed intestinal mucosa in patients with IBD has the same capacity to transport and oxidize butyrate as mucosa from healthy patients, thus refuting the theory that an inherent deficiency in butyrate utilization is the basis for IBD [67]. Rather, butyrate deficiency appears to be a consequence of intestinal inflammation, instead of its cause. Furthermore, MCT1 is downregulated in inflamed intestinal tissue, thus decreasing intracellular butyrate availability and defeating the purpose of butyrate supplementation during active inflammation [68,69].

Nevertheless, butyrate does modulate intestinal inflammation, and dysbiosis in IBD may contribute to disease severity [70,71]. Butyrate mediates intestinal inflammation predominantly through binding of free fatty acid receptors (FFAR), a family of GPCR present on the epithelial cell membrane [72,73,74]. GPR41 and GPR43, also known as FFAR3 and FFAR2, respectively, are activated by SCFA in the colon and small intestine, and lead to production of anti-inflammatory cytokines including IL-18 [75,76].

Additionally, butyrate has been associated with increased production of regulatory T cells (Treg), which modulate the severity of immune response [77,78]. Butyrate-mediated HDAC inhibition enhances expression of the Foxp3 locus via enhanced histone H3 acetylation, which is central to Treg differentiation [78]. In an animal model of colitis, mice receiving oral butyrate supplementation demonstrated less severe inflammation and lower levels of histone H3 acetylation in epithelial cells than untreated controls [79,80].

Overall, the local anti-inflammatory effects of butyrate appear to render epithelial cells tolerant of commensal bacteria, and additionally confer a protective role against unchecked inflammation [81,82]. There is a growing body of evidence to suggest that in IBD, epithelial and immune cells exhibit resistance to butyrate’s anti-inflammatory function. Immune cells from IBD patients require higher concentrations of butyrate to downregulate pro-inflammatory cytokines than those from healthy patients [83], and butyrate induces greater expression of anti-inflammatory genes in intestinal samples of healthy patients compared to patients with ulcerative colitis (UC) [83,84]. Though this attenuated responsiveness may be due in part to the reduced cellular uptake of butyrate observed in inflamed epithelial cells, this effect also appears to be mediated by GPCR signaling that occurs irrespective of intracellular butyrate concentration [80].

Therapeutic application of butyrate and targeting of butyrate’s downstream targets has yet to prove fruitful in the treatment of IBD. Small scale trials have demonstrated the safety and tolerability of oral butyrate for patients with UC and Crohn’s disease, but have failed to show significant improvement in symptoms or intestinal histology, which may be due to presence of butyrate-resistant phenotype in IBD patients [85,86]. Early investigation into the use of probiotics in intestinal inflammation has shown some promise in attenuating epithelial cell resistance to butyrate’s anti-inflammatory effects [87]. Further research into this nuanced interaction will be needed to develop translational therapies exploiting butyrate’s protective function in intestinal inflammation.

### 4.2. Tolerogenic Effects of Butyrate in the Intestinal Epithelium

As mentioned above, butyrate has been linked to enhanced Treg production, which, by modulating the epithelial immune response in the face of perpetual environmental and food antigen exposure, is critical for prevention of food allergy [77]. The pathways by which butyrate and other SCFA increase immunologic tolerance are varied. In addition to enhanced T cell differentiation via butyrate’s HDAC properties, dietary fiber and butyrate promote Treg production through increased activity of tolerogenic epithelial dendritic cells [88]. Butyrate-induced dendritic cell tolerance in the epithelium is associated with mucosal IgA production, which provides a critical line of immunologic defense in the intestine [89,90]. Furthermore, there is evidence that butyrate may inhibit mast cell degranulation in the intestinal mucosa, thereby limiting the release of circulating inflammatory mediators [91].

Together, butyrate’s effects on the immunologic milieu of the epithelium have consequences for the development and resolution of food allergy. Research into cow’s milk allergy suggests that composition of the microbiome in early life is associated with allergic phenotype, and that enrichment of butyrate-producing bacteria and higher fecal butyrate levels predict tolerance [59,60].

The mechanisms by which butyrate protects against food allergy are closely related to the barrier function of the epithelium, which comprises one of the intestine’s greatest immunologic defenses (reviewed by Pardo-Camacho, et al. [92]). Specifically, inhibition of mast cell degranulation and Treg proliferation precludes cytokine and histamine-induced pathways toward increased intestinal permeability [93,94,95,96]. Through its impact on epithelial oxygen consumption, butyrate also results in stabilization of hypoxia inducible factor, a transcription factor involved in maintenance of epithelial tight junctions [97,98]. Moreover, butyrate’s activity as an HDAC inhibitor enhances expression of antimicrobial peptides and mucin, which serve to protect the intestinal epithelium from pathogens [99,100,101].

## 5. SCFA and the Gut–Brain Axis

Growing evidence linking microbiome composition and alteration to systemic disease has elucidated the immunomodulatory role of circulating SCFA on extraintestinal organ systems. Of particular interest is the bidirectional interaction between intestinal homeostasis and nervous system function, including memory, mood and neuronal recovery after injury. As mentioned previously, systemic absorption of SCFA from the intestine is minimal, with butyrate exhibiting lower bloodstream concentrations than propionate and acetate [3]. Thus, despite permeability of SCFA across the blood–brain barrier, the actual concentration of these molecules in the brain is negligible [102,103]. Rather than direct activation of neuronal receptors, a more likely proposed mechanism for SCFA influence on neural processes is through regulation of neuroinflammation [104,105].

SCFA activation of GPR43 and GP41 receptors on immune precursor cells leads to neutrophil activation and recruitment to sites of infection, thus playing an important role in mounting an inflammatory response, both within and beyond the intestinal epithelium [66,106]. In the brain, there is evidence that butyrate downregulated lipopolysaccharide-induced neuroinflammation in microglia, a specialized population of immune cells important for neuroprotection and neural remodeling [104]. Indeed, microbiome composition and SCFA levels have been implicated in the pathogenesis of substance use disorders, Parkinson’s disease and depression [107,108,109].

Gut–brain crosstalk in disease pathogenesis is also evident in nervous system responsiveness to SCFA-induced gut hormone secretion [110]. Butyrate and propionate lead to intestinal gluconeogenesis via FFAR activation, which in turn leads to central nervous system sensing of portal glucose levels and metabolic regulation [105]. Additionally, butyrate stimulates enteroendocrine cells in the intestinal epithelium to release hormones, including glucagon-like peptide 1 and serotonin, which act on neuronal receptors to modulate neuronal processes in normal and injured states [111,112,113,114].

## 6. Conclusions

Butyrate serves as a fundamental, yet versatile molecular link between the microbiome and the host intestine. First, colonocyte and enterocyte oxidation pathways utilize butyrate as cellular fuel, a basic cellular need. However, this simple function underlies the numerous ways the epithelium has adapted to interact with it, both physiologic and pathologic. Indeed, the epithelial crypt morphology demonstrates how the intestine has evolved to conserve rapid glycolytic energy extraction for undifferentiated stem cells, and to protect proliferating cells from direct effects from the microbiome. Furthermore, intracellular accumulation of butyrate has evolved as a protective mechanism against carcinogenesis. The nuances of transepithelial butyrate transport and activation of membrane-bound receptors in colon cancer and intestinal inflammation reveal an impressive level of epithelial cell auto-regulation in response to the microbial milieu. Continued research is warranted to develop methods for measurement and regulation of butyrate activity in the intestinal epithelium, as these tools have the potential to expand our capacity to treat, prevent and evaluate intestinal disease.

## Figures and Tables

**Figure 1 cells-10-01775-f001:**
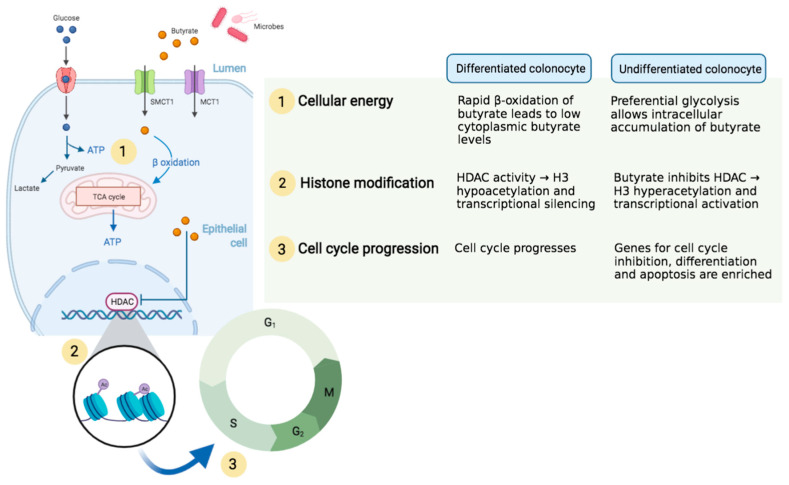
Distinct metabolic pathways for cellular energy in differentiated and undifferentiated colonocytes are responsible for ‘the butyrate paradox’. Microbially-derived butyrate in the colonic lumen is transported intracellularly by SMCT1 and MCT1, among other membrane-bound transport proteins. The fate of cytoplasmic butyrate depends on the cell’s state of differentiation. (1) In differentiated colonocytes, butyrate is rapidly oxidized and utilized for cellular energy production. In undifferentiated colonocytes, such as neoplastic cells, cellular fuel is preferentially derived from glycolytic pathways, leading to intracellular butyrate accumulation. (2) Butyrate is an HDAC inhibitor, which results in hyperacetylation of histone H3 with consequent chromatin relaxation and expression of genes otherwise silenced by HDAC activity. (3) Specifically, butyrate-induced histone modification is associated with transcriptional activation of genes involved in cell cycle inhibition and apoptosis.

**Figure 2 cells-10-01775-f002:**
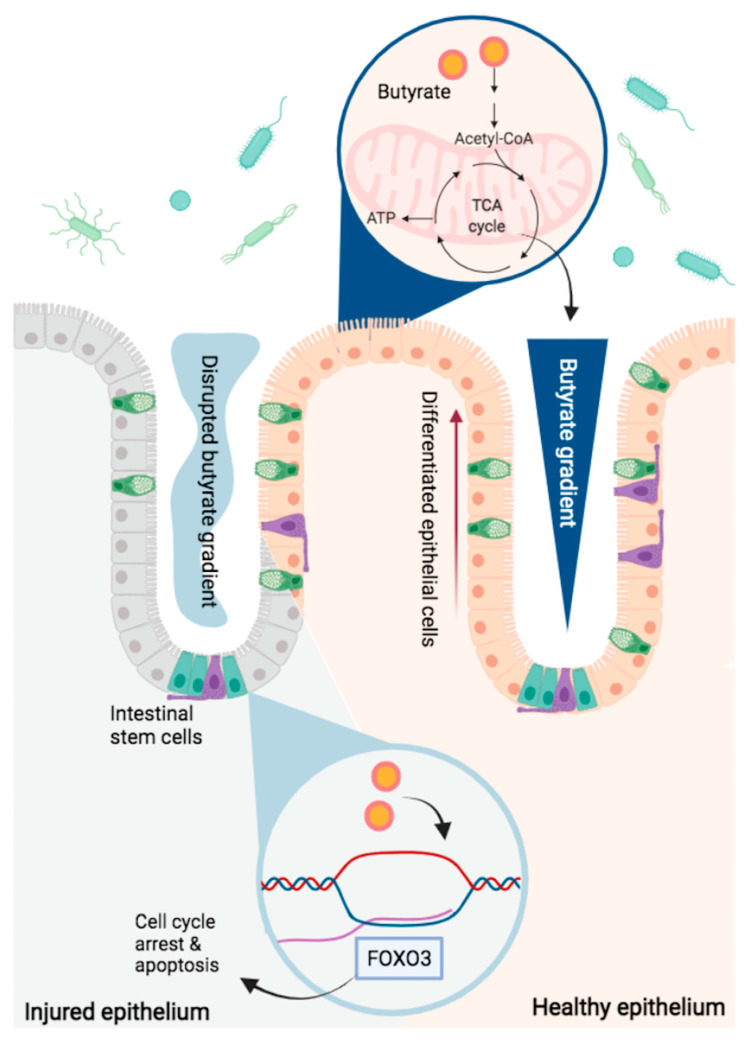
The epithelial crypt in mice facilitates production of a butyrate gradient which is necessary to maintain the proliferative potential of intestinal stem cells. Butyrate is an abundant energy source for surface level epithelial cells in the mammalian intestine. Rapid intracellular transport and oxidation by these cells produces a butyrate gradient, with significant depletion of butyrate levels at the crypt base. Disrupted crypt morphology, as in the case of mucosal injury, allows butyrate levels to rise at the crypt base, where it induces antiproliferative activity in intestinal stem cells, mediated by the transcription factor Foxo3.

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
