# Peer review of "Butyrate and the Intestinal Epithelium: Modulation of Proliferation and Inflammation in Homeostasis and Disease"

_cells, 2021, doi:10.3390/cells10071775_

Round 1

Reviewer 1 Report

Interesting review on the role of butyrate in the physiology and pathophysiology of intestinal epithelium. However, in my opinion, this review lacks clinical application of butyrate. It is interesting, because both authors work in the Department of Surgery.

Please include the role of butyrate in gut tolerance and  in the prevention of food allergies

What is the effect of butyrate on the function of intestinal barrier ??

Only 20 bacterial species are able to produce SCFA.-which bacteria play a fundamental role in the production of SCFA ?

What is the effect of SCFA on microbiota brain gut axis ??

Author Response

Response to Reviewer # 1

Interesting review on the role of butyrate in the physiology and pathophysiology of intestinal epithelium. However, in my opinion, this review lacks clinical application of butyrate. It is interesting, because both authors work in the Department of Surgery.

We thank the reviewer for their careful consideration and feedback. While we agree that discussion of the clinical applications of butyrate is limited in our review, we feel this reflects a lack of clinical research. We have referenced clinical studies where applicable (Lines 305-311) and have otherwise alluded to the potential clinical implications of current knowledge. We hope that further translational studies of the microbiome and butyrate will shed light on how these may be manipulated for clinical use.

Please include the role of butyrate in gut tolerance and in the prevention of food allergies

Thank you for this suggestion. We have modified our review to include a section on the role of butyrate in immune tolerance and food allergy (Section 4.2).

What is the effect of butyrate on the function of intestinal barrier ??

We appreciate this point of feedback and have accordingly included a discussion describing butyrate’s role in intestinal barrier integrity (Lines 362-372)

Only 20 bacterial species are able to produce SCFA.-which bacteria play a fundamental role in the production of SCFA ?

Thank you for highlighting this omission in our text. We have added mention of the bacterial species that are most important for butyrate production in the colon (Lines 54-60).

What is the effect of SCFA on microbiota brain gut axis ??

Initially, we had not included discussion of the gut-brain axis as we considered this beyond the scope of our review. Upon consideration of the reviewer’s comments, we believe that a summary of how butyrate-producing bacteria and SCFA play a role in neuronal function is relevant to our discussion of the immunomodulatory effects of butyrate. As such, we have added section 5 (line 375) to review this topic.

Reviewer 2 Report

Shah and Cowles wrote a well structured, easy to follow small Review about butyrate and its impact on the intestinal epithelium, with focus on proliferation/cell differentiation and inflammation.

Many reviews were already written on butyrate, its production and its potential health benefits. This manuscript offers a slightly new point of view by developing the butyrate paradox and the importance of butyrate concentration gradient in intestinal crypts and its importance in the coevolution of microbiome and host. This make the manuscript of interest for readers.

Still I would have some suggestions to try to improve the manuscript. 

First there is a strong discrepency between point 3 (proliferation) and 4 (inflammation). I would welcome some more molecular insights of the action of butyrate on inflammation. Line 208, autors mention "production of anti-inflammatory cytokines". If this is true, but it would gain being developed: which one, how? (for example: binding to GPR109A leads to IL18 production in IEC but also to activation of inflammasome pathway in macrophages and dendritic cells, which leads to differentiation of T cells (Treg and IL10 producing ones)...) Moreover anti-inflammation effects might be also related to the indirect effect of butyrate on oxidative stress. Considering the whole intestinal barrier as a way to avoid inflammation, the positive effects of butyrate on the expression of genes like MUC2, TFFs, Tigh-Junction proteins, antimicrobial peptides... should also be mentioned. At least a reference to another review or so should appear. 

Also the §4 is strongly oriented to clinical issues (IBD, UC) with mention of human studies. This is good, but why not doing the same for the §3.2 on cancer? There might be less human studies but many animal models, and epidemiologic studies (at least with fibers). Here I would enjoy some more references in whole organisms.

Beside this 2 "bigger" points, I have following minor points:

-title should be adapted as only cancer and inflammation are really reviewed.

-abstract: something went wrong with the "beta" L13

-L46: ...fermentation MAINLY of dietary starch/FIBERS. Indeed starch are a kind of fiber, butyrate can be produced from other fiber types. moreover it can be produced through protein fermentation (even if in much lower quantities).

-§2 is named "production and...". I recognise that it was not the point of this manuscript to review the production (which microorganisms, which pathways...). Still a short mention (like many gram + bacteria, two main pathways...) and ideally a reference to another review (like Autors did L59), where details thereabout could be find, would be good here.

-Fig1: top bottom right: I would add "differenciation" between inhibition and apoptosis. This would fit to the text of the manuscript.

-L122: "downstream": I am not so sure what you mean. Would not be "lowering" or "reduction of " clearer (as Bcl-2 is an anti-apoptotic factor)

-fig2: the gradient of differentiated IEC should be nearer the butyrate gradient (rather on the right) otherwise it could be misunderstood. A mention of apoptosis in the grafic (enhanced through FOXO3) would also be helpful.

-§3.3: I would like autors to shortly explained why in this § butyrate inhibition of proliferation take place through FOXO3 whereas in §3.2 it works through Bax and Bak genes. Both cancer cells and stem cells use mainly glucose as fuel. What are the differences between these 2 cell types, that explain the different pathway of butyrate (even if the effect, ie less proliferation) is the same. If no explanation/hypotheses yet are known, i would just appreciate mentionning it. 

-L160: please replace "diverse" by "butyrate producing", as you cannot conclude about anything else with the shown data/references.

-L245: accumulation OF butyrate

Thanks in advance for taking my comments into account!

Best regards 

Author Response

Reviewer #2

Shah and Cowles wrote a well structured, easy to follow small Review about butyrate and its impact on the intestinal epithelium, with focus on proliferation/cell differentiation and inflammation.

Many reviews were already written on butyrate, its production and its potential health benefits. This manuscript offers a slightly new point of view by developing the butyrate paradox and the importance of butyrate concentration gradient in intestinal crypts and its importance in the coevolution of microbiome and host. This make the manuscript of interest for readers.

Still I would have some suggestions to try to improve the manuscript.

First there is a strong discrepency between point 3 (proliferation) and 4 (inflammation). I would welcome some more molecular insights of the action of butyrate on inflammation. Line 208, autors mention "production of anti-inflammatory cytokines". If this is true, but it would gain being developed: which one, how? (for example: binding to GPR109A leads to IL18 production in IEC but also to activation of inflammasome pathway in macrophages and dendritic cells, which leads to differentiation of T cells (Treg and IL10 producing ones)...) Moreover anti-inflammation effects might be also related to the indirect effect of butyrate on oxidative stress. Considering the whole intestinal barrier as a way to avoid inflammation, the positive effects of butyrate on the expression of genes like MUC2, TFFs, Tigh-Junction proteins, antimicrobial peptides... should also be mentioned. At least a reference to another review or so should appear.

Thank you for your constructive feedback. We agree with these points and have added a more in-depth description of the mechanisms by which butyrate regulates inflammation, including a discussion of Treg proliferation, dendritic cell tolerance, and the intestinal barrier (see lines 285-291, and lines 316-373).

Also the §4 is strongly oriented to clinical issues (IBD, UC) with mention of human studies. This is good, but why not doing the same for the §3.2 on cancer? There might be less human studies but many animal models, and epidemiologic studies (at least with fibers). Here I would enjoy some more references in whole organisms.

We agree with the reviewer’s thoughtful assessment and have added a few sentences (Lines 155-162 describing animal models and human studies that relate to the clinical applicability of butyrate and colon cancer. Additionally, we had previously cited various epidemiological studies in humans linking fiber intake to colon cancer (Line 88-91).

Beside this 2 "bigger" points, I have following minor points:

-title should be adapted as only cancer and inflammation are really reviewed.

We have adjusted the title to more accurately reflect the focus of the review.

-abstract: something went wrong with the "beta" L13

This error has been corrected.

-L46: ...fermentation MAINLY of dietary starch/FIBERS. Indeed starch are a kind of fiber, butyrate can be produced from other fiber types. moreover it can be produced through protein fermentation (even if in much lower quantities).

Thank you for identifying this inaccuracy. The sentence has been revised to be more precise. (Line 55)

-§2 is named "production and...". I recognise that it was not the point of this manuscript to review the production (which microorganisms, which pathways...). Still a short mention (like many gram + bacteria, two main pathways...) and ideally a reference to another review (like Autors did L59), where details thereabout could be find, would be good here.

We agree with this suggestion and have revised the first paragraph of Section 2.

-Fig1: top bottom right: I would add "differenciation" between inhibition and apoptosis. This would fit to the text of the manuscript.

This change has been made in the figure.

-L122: "downstream": I am not so sure what you mean. Would not be "lowering" or "reduction of " clearer (as Bcl-2 is an anti-apoptotic factor)

This wording has been altered for clarity (Line 152).

-fig2: the gradient of differentiated IEC should be nearer the butyrate gradient (rather on the right) otherwise it could be misunderstood. A mention of apoptosis in the grafic (enhanced through FOXO3) would also be helpful.

These changes have been made in Figure 2.